# Hepatitis C antibody screening and determinants of initial and duplicate screening in the baby boomer patients of six urban primary care clinics

**Dagan Coppock**[1¤a]*, **Edgar Chou**[2¤b], **Edward Gracely**[3], **Robert Gross**[4], **Dong Heun-Lee**[1]

**1** Department of Medicine, Division of Infectious Diseases and HIV Medicine, Drexel University College of Medicine, Philadelphia, Pennsylvania, United States of America, **2** Department of Medicine, Division of General Internal Medicine, Drexel University College of Medicine, Philadelphia, Pennsylvania, United States of America, **3** Dornsife School of Public Health, Drexel University, Philadelphia, Pennsylvania, United States of America, **4** Perelman School of Medicine, University of Pennsylvania, Philadelphia, Pennsylvania, United States of America

¤a Current address: Department of Medicine, Division of Infectious Diseases, Sidney Kimmel Medical College, Thomas Jefferson University, Philadelphia, Pennsylvania, United States of America
¤b Current address: Department of Medicine, Division of Internal Medicine, Sidney Kimmel Medical College, Thomas Jefferson University, Philadelphia, Pennsylvania, United States of America
* dagan.coppock@jefferson.edu

**Data Availability Statement:** All relevant data are within the manuscript and its Supporting Information files.

## Abstract

### Introduction

In 2012, the Centers for Disease Control and Prevention released updated guidelines recommending universal, one-time hepatitis C virus screening for all individuals born between 1945 and 1965. Prior to the implementation of these guidelines, testing rates were inappropriately low, but unnecessary duplicate antibody testing was also problematic. In the era of increased efforts to screen "baby boomers", the prevalence and social determinants of initial and duplicate hepatitis C testing have not been well described.

### Methods

A hepatitis C screening program was implemented at six urban primary care clinics affiliated with Drexel University College of Medicine. Data was collected regarding the screening patterns in these clinics. Annual screening rates for the program were assessed. Multivariate logistic regression analyses were used to examine the association of demographic variables and the outcomes of subjects having ever been tested and subjects having received duplicate testing.

### Results

Following the implementation of the program, the screening rate increased from 16% in the first year of analysis to 82% in the final year of analysis. Of the 6,717 patients screened, 1,207 had duplicate testing, of which 14% had inappropriate duplicate antibody screening.

**Funding:** Dr. Dagan Coppock received support from the the National Institute of Allergy and Infectious Diseases T32 AI055435 Dr. Edgar Chou received the Gilead Sciences FOCUS grant.

**Competing interests:** The authors have declared that no competing interests exist.

African Americans and Asian patients had a higher odds of being screened. Patients with public insurance had a higher odds of duplicate screening.

## Conclusions

In the setting of an aggressive hepatitis C screening program, high testing rates may be attained in a target population. However, inappropriate duplicate antibody testing rates may be high, which may be a burden in resource-limited settings.

## Introduction

In 2012, the Centers for Disease Control and Prevention (CDC) released updated guidelines recommending one-time hepatitis C virus (HCV) screening for all individuals born between 1945 and 1965—the so-called "baby boomer" birth-cohort [1]. To reflect these guidelines, the state of Pennsylvania enacted the Hepatitis C Screening Act, which requires that "each individual born between the years of 1945 and 1965, who receives primary care services in an outpatient department of a hospital, health care facility or physician's office shall be offered a hepatitis C screening test" [2].

Prior to the initiation of these guidelines, waste in HCV screening has been a concern. Between 2006 and 2010, inappropriate duplicate antibody screening cost New York City an estimated $14 million [3]. An evaluation of HCV screening in United States Veterans Affairs Hospitals demonstrated similar duplication patterns prior to the updated guidelines [4]. Despite the availability of previous testing in the electronic health record (EHR), forty percent of Veterans who had an initial positive screen had inappropriate duplicate testing [4]. In the era of broader HCV screening, the Centers for Medicare and Medicaid Services have called on providers to avoid inappropriate duplicate HCV screening so as to avoid waste and ensure coverage [5]. However, the burden of duplicate testing, both redundant and potentially appropriate, in baby boomers has yet to be described.

To address the CDC's 2012 recommendations, Drexel University College of Medicine (DUCOM) created the C for Cure program to spearhead HCV screening efforts in baby boomers at six of its urban primary care clinics. These clinics provide care for a medically-underserved patient population, many of whom require public insurance, with significant barriers to HCV screening [6, 7]. In this study we examine the results of the C for Cure's screening program, including overall testing rates, patterns of duplicate testing, and social determinants for overall and duplicate testing.

## Materials and methods

### Study design

The study was conducted as a retrospective cohort study.

### Study population

Birth-cohort data were extracted in deidentified datasets for all patients born between January 1, 1945 and December 31, 1965. To be considered eligible for the study, patients were required to be seen at least once at one of six DUCOM primary care sites between January 1, 2012 and July 31, 2017. This period was chosen given the implementation of updated CDC guidelines in 2012. Further, to be included in the analysis of a specific year, patients were required to be

seen at least once during the year being evaluated. Patients were deemed ineligible for analysis for a given year if they were not seen during that year. Covariates of interest, linked as structured data to deidentified birth-cohort patients, were collected as a part of the data extraction from the DUCOM EHR. Covariates collected included birth year, gender, race, insurance status, practice location, and HCV screening and screening duplication results as defined below. Age was defined as patient age at the midpoint of the study.

Throughout the screening process, patients were followed by the C for Cure team, which included providers, nurses and patient navigators. The team provided patients with guidance regarding follow-up plans and treatment if warranted by the testing results. EHR decision support and provider education were included to help augment screening efforts at each of the six clinics.

### Data source

All testing and demographic data were collected from the DUCOM outpatient EHR as deidentified data in the C for Cure database.

### Definitions

HCV screening was defined as the completion of an HCV serum antibody test, regardless of the result. Duplicate testing was defined as repeat HCV antibody testing performed at any time point subsequent to the first recorded screening test. Inappropriate duplicate antibody screening was defined as antibody screening following an initial positive antibody screen. Potentially appropriate duplicate antibody screening was defined as repeat antibody testing for patients who had an initial negative HCV serum antibody screen. These tests were deemed potentially appropriate as these patients may have had a secondary ongoing risk—that is, a risk other than being birth-cohort patients—that may predispose them to seroconversion [8].

### Statistical analysis

Screening rates were calculated based upon eligible patients seen in a given year. Of those eligible patients, screening was assessed for both newly-screened and previously-screened individuals. Counts and simple percentages were performed for pooled testing and demographic data from all years of the study. Multivariate logistic regressions with clinics controlled as clusters were performed to examine the association of available demographic variables and the outcomes of having ever received HCV screening as well as having ever received duplicate HCV screening (alpha 0.05). All analyses were performed with Stata (Version 16.1, StataCorp, College Station, Texas).

### Ethics

Approval of this project was provided by the Drexel University College of Medicine Institutional Review Board (Protocol Approval # 1702005228). A waiver of consent and waiver of Health Insurance Portability and Accountability Act authorization were provided.

### Results

Baseline characteristics of the patient population eligible to be screened are summarized in Table 1. 11,598 total eligible patients were evaluated for the study. Screening was largely uniform across age cohorts. Those screened were predominantly female, African American, and those with public insurance. Full data was available for all covariates with the exception of insurance, for which 272 patients had an unavailable insurances status.

**Table 1. Baseline characteristics of eligible patients.**

| Birth Cohort Patients Seen 2012–2017 | Population Size N (% of Total) |
|---|---|
| **Total** | 11,598 (100) |
| **Gender** | |
| Male | 4627 (39.9) |
| Female | 6971 (60.1) |
| **Age cohort (years old)** | |
| 50–54 | 3173 (27.3) |
| 55–59 | 3263 (28.1) |
| 60–64 | 2758 (23.7) |
| 65–70 | 2404 (20.7) |
| **Race** | |
| White | 4625 (39.9) |
| Black | 5998 (51.7) |
| Asian | 217 (1.9) |
| Others | 758 (6.5) |
| **Insurance** | |
| Private | 4039 (34.8) |
| Public | 7287 (62.8) |
| Unavailable | 272 (2.3) |
| **Practices** | |
| Clinic 1 | 2353 (20.2) |
| Clinic 2 | 2381 (20.5) |
| Clinic 3 | 2715 (23.4) |
| Clinic 4 | 2420 (20.8) |
| Clinic 5 | 987 (8.5) |
| Clinic 6 | 742 (6.4) |

Cumulative screening rates for all clinical sites are described, by year, in Fig 1. Screening rates are characterized in the figure based upon the eligible pool of patients seen during a given year, including those who were newly-tested, previously-tested, or not tested. The cumulative HCV screening rate increased from 16% in 2012 to 82% in 2017. In 2012, previously-tested patients accounted for 11% of the total pool of eligible patients, while the newly-tested patients accounted for 5% of eligible patients. A gradual increase was seen in both the newly-tested and previously-tested patients. As of 2017, previously-tested patients made up 66% of the total eligible patients, while newly-tested patients made up 16% of the pool.

Cumulative screening results and duplication patterns are described in Fig 2. During the study period, a total of 7927 screening tests were performed. Among the 6,717 patients screened, 1,208 (18%) had duplicate testing, only three of which had a third screening test. Of those who had duplicate testing, 170 (14%) had inappropriate duplicate antibody screening. 1,037 patients had duplicate screening that was potentially appropriate. Of those patients who had potentially appropriate duplicate testing, three seroconverted to a positive result on duplicate testing.

The outcomes of general and duplicate screening were analyzed through multivariate logistic regression, adjusted for clustering by clinic, to explore demographic determinants of each (Tables 2 and 3). Prior to adjustment for clustering by clinic, race, type of insurance and clinic location were all associated with patients having ever been screened for HCV. After adjustment for clustering by clinic, African American and Asian patients had a significantly higher

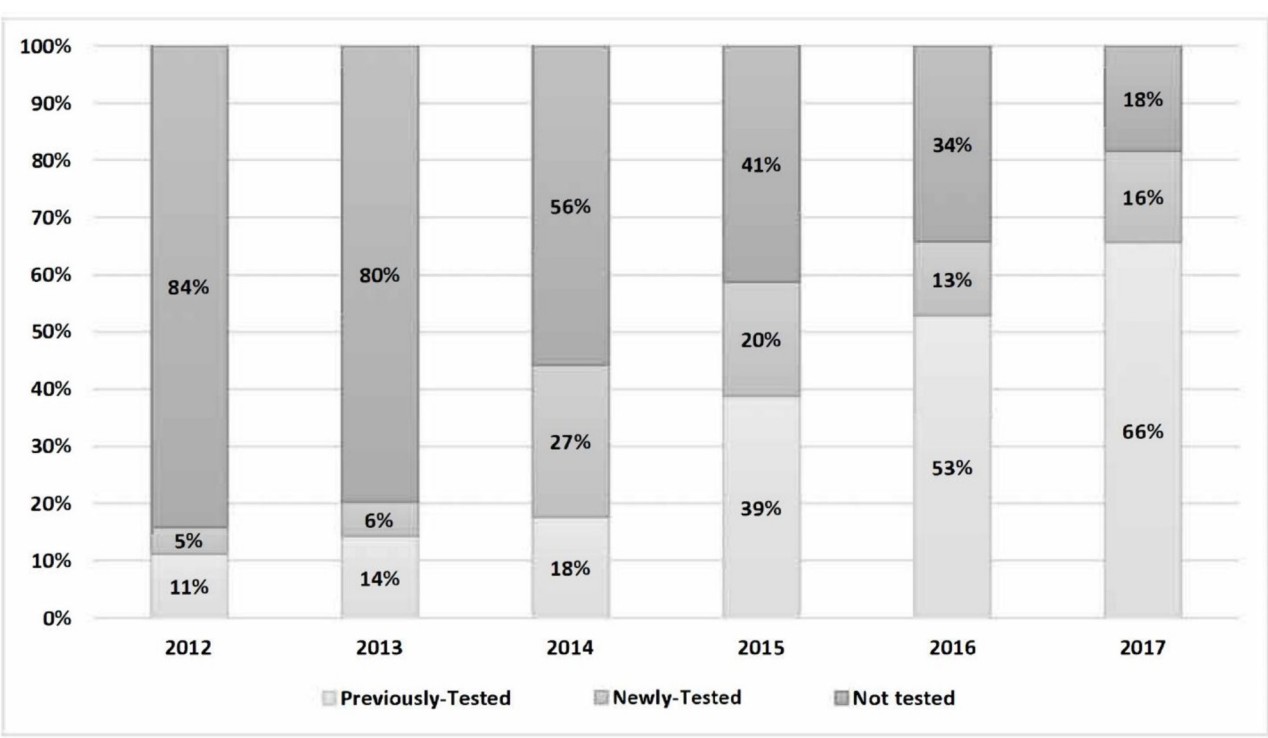

**Fig 1. Screening rates for eligible patients, by year.**

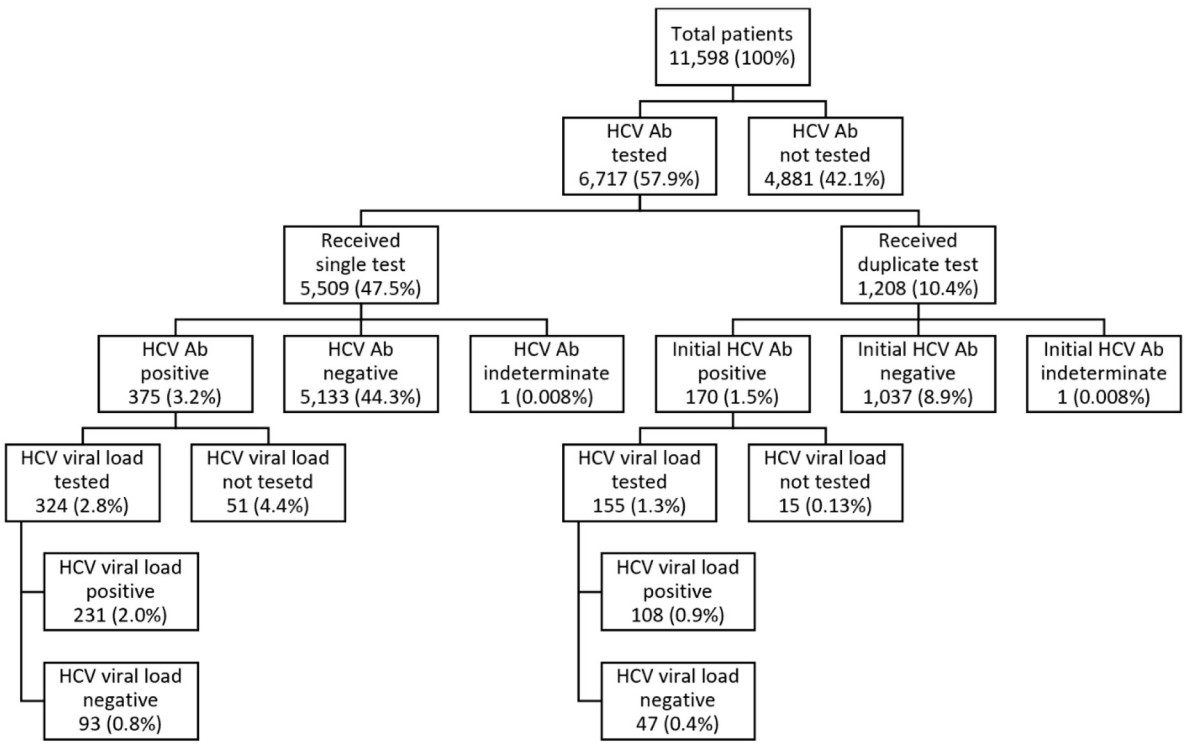

**Fig 2. Screening and duplication patterns.** HCV, hepatitis C virus; Ab, antibody.

**Table 2. Multivariate logistic regression for ever receiving HCV screening, clustered by clinic.**

| Patients Seen From 2012–2017 | HCV Ever- Tested N (%) | HCV Never- Tested N (%) | Odds Ratio (95% CI) | p-value |
|---|---|---|---|---|
| **Gender** | | | | |
| Male | 2706 (58.5) | 1921 (41.5) | 1 | |
| Female | 4011 (57.5) | 2960 (42.5) | 0.89 (0.67–1.18) | 0.413 |
| **Birth Year** | | | | |
| 1945–1955 | 2936 (56.9) | 2226 (43.1) | 1 | |
| 1956–1965 | 3781 (58.7) | 2655 (41.3) | 1.06 (0.95–1.18) | 0.285 |
| **Race** | | | | |
| White | 2383 (51.5) | 2242 (48.5) | 1 | |
| Black | 3830 (63.9) | 2168 (36.1) | 1.71 (1.27–2.29) | <0.001 |
| Asian | 137 (63.1) | 80 (36.9) | 1.68 (1.25–2.26) | <0.001 |
| Others/unknown | 354 (47.7) | 388 (52.3) | 0.91 (0.65–1.28) | 0.598 |
| **Insurance** | | | | |
| Private | 3776 (57.7) | 2763 (42.3) | 1 | |
| Public | 2856 (59.7) | 1930 (40.3) | 0.95 (0.77–1.18) | 0.639 |

HCV, hepatitis C virus; CI, confidence interval

odds for ever being screened. Prior to adjustment for clustering, age, race, type of insurance and clinic location were all associated with duplicate testing. After adjustment, Asian patients and those with public insurance had a significantly higher odds of duplicate screening.

## Discussion

Despite expanded CDC guidelines for universal HCV screening, which now includes all adults [9], screening remains a challenge for higher risk patients with less access to care [7, 10–13]. Nonetheless, there have been successful screening programs described for such populations [14, 15]. Through the DUCOM C for Cure program, 58% of 11,598 baby boomer patients were screened for HCV during the study period. When we reviewed patients screened by year,

**Table 3. Multivariate logistic regression for duplicate HCV screening, clustered by clinic.**

| Patients Seen From 2012–2017 | HCV-Tested More Than Once N (%) | HCV-Tested Once N (%) | Odds Ratio (95% CI) | p-value |
|---|---|---|---|---|
| **Gender** | | | | |
| Male | 456 (16.9) | 2250 (83.1) | 1 | |
| Female | 751 (18.7) | 3620 (81.3) | 1.06 (0.88–1.28) | 0.510 |
| **Birth Year** | | | | |
| 1945–1955 | 485 (16.5) | 2451 (835) | 1 | |
| 1956–1965 | 772 (19.1) | 3059 (80.9) | 1.15 (0.95–1.41) | 0.158 |
| **Race** | | | | |
| White | 259 (10.9) | 2124 (89.4) | 1 | |
| Black | 862 (22.5) | 2968 (77.5) | 1.62 (0.83–3.15) | 0.157 |
| Asian | 23 (16.8) | 114 (83.2) | 2.17 (1.70–2.78) | <0.001 |
| Others/unknown | 293 (82.8) | 293 (82.8) | 1.71 (1.05–2.78) | 0.030 |
| **Insurance** | | | | |
| Private | 561 (14.9) | 3215 (85.1) | 1 | |
| Public | 630 (22.1) | 2226 (77.9) | 1.40 (1.01–1.93) | 0.042 |

HCV, hepatitis C virus; CI, confidence interval

the HCV screening rate increased from 16% screened previously or in 2012 to 82% in 2017. In 2017, not only had most patients in the practices already been tested, but almost half of the remaining untested patients were tested in that year.

Our findings illustrate that a high rate of HCV screening can be achieved with the implementation of a comprehensive program, despite our population's previously described barriers to care [6]. Previous data have supported the use of such programs. HCV continuum of care teams that include patient navigators improve screening and linkage-to-care [7, 16–18]. Furthermore, physician education and clinical support, including EHR-based tools, are likewise critical to improving testing rates [19–24].

Though the program was successful, we noted a significant amount of inappropriate duplicate HCV antibody screening, defined as repeat antibody testing in patients whose initial antibody tests were positive. Fourteen percent of all duplicate screening tests were inappropriate. For these patients, providers concerned about ongoing risk could have evaluated that risk of infection more appropriately through HCV viral load testing. The 1,037 remaining duplicates were potentially appropriate as the initial screening for these patients was negative. However, this study did not explore whether the potentially appropriate duplicate tests were truly appropriate re-screens or were the consequence of provider error. For example, patients with previously negative screens may have had duplicate testing for an appropriate reason, such as new or ongoing injection drug use. Alternatively, inappropriate duplicate antibody testing may have occurred if providers were unaware of previous screening or were unknowledgeable of the clinical appropriateness of re-screening.

Our analyses suggest that, among our population, there were significant social determinants for both having ever been tested as well as duplicate testing. For having ever been screened, race was a significant determinant of testing, particularly for African American and Asian patients. This may be reflective of more aggressive screening due to known racial disparities associated with HCV prevalence and related mortality [25, 26]. In regards to duplicate testing, race also appeared to remain a significant determinant for Asian patients, though not for African American patients. Further, having public insurance was significantly associated with duplication. This may be due to the City of Philadelphia's increased efforts at screening during the study period [2] as well as its liberalized coverage policies for HCV treatment in publicly insured patients [27, 28].

There are limitations to this study. The study only collected data on baby boomers without the collection of data on other comorbidities. This may be problematic for generalizability, particularly as it relates to baby boomers with more complex social and medical histories. For example, injection drug use remains prevalent in the United States and is an important risk for HCV acquisition [29, 30]. The CDC cites other risk factors for HCV infection, including HIV infection, incarceration, chronic hemodialysis, receipt of distant transfusions, and needlestick injuries [31]. Though not currently cited by the CDC, unprotected anal receptive sex among men who have sex with men has also been associated with an increased risk for HCV infection [32]. Future studies will need to address these risk factors and how they impact repeat HCV screening. Further, we hope to expand our evaluation beyond baby boomers to reflect universal adult screening recommendations as introduced by the CDC this year [9].

## Conclusions

In summary, this study describes the potential for inappropriate duplicate HCV antibody screening in the setting of updated CDC guidelines. For a large-scale program, such as C for Cure, inappropriate duplicate antibody testing may occur, which should prompt efforts to

improve resource stewardship with directed repeat testing, using the appropriate modality, in patients with ongoing risk.

## Supporting information

**S1 Data.**
(XLSX)

## Author Contributions

**Conceptualization:** Edgar Chou, Dong Heun-Lee.

**Data curation:** Edward Gracely, Dong Heun-Lee.

**Formal analysis:** Edward Gracely.

**Investigation:** Dagan Coppock, Dong Heun-Lee.

**Methodology:** Edgar Chou, Dong Heun-Lee.

**Writing – original draft:** Dagan Coppock.

**Writing – review & editing:** Dagan Coppock, Edgar Chou, Edward Gracely, Robert Gross, Dong Heun-Lee.

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
