## [Decision Letter · Decision Letter 0]

31 Mar 2020

PONE-D-20-03702

Screening Baby Boomers for Hepatitis C in Urban Primary Care Clinics: Does Duplicate Testing Make a Difference?

PLOS ONE

Dear Dr. Coppock,

Thank you for submitting your manuscript to PLOS ONE. After careful consideration, we feel that it has merit but does not fully meet PLOS ONE’s publication criteria as it currently stands. Therefore, we invite you to submit a revised version of the manuscript that addresses the points raised during the review process.

The manuscript was evaluated by peer-review team and I found some interesting results however, we have major concerns with the study aims, results and presentation. Unfortunately, the current problem with testing baby boomers and duplicating the test was not presented clearly. Besides, the aim of the study looks confusing. Reading the manuscript title, I expected that duplicate testing is a new strategy for screening of Hep C patients and the study is going to evaluate this strategy. Reading introduction and aim, I thought the duplicate testing is something unexpected in the primary care clinics but it is not elucidated why it cannot be prevented while they have EHR to document the HCV testing. We commonly recommend HCV retesting in populations with ongoing risky behaviors such as PWIDs however as this study found baby boomers are not an appropriate population for HCV retesting and in HCV retesting, one was diagnosed as newly contracted HCV and spontaneously resolved the infection resulted in finding no new case for treatment. 

We would appreciate receiving your revised manuscript by May 15 2020 11:59PM. To enhance the reproducibility of your results, we recommend that if applicable you deposit your laboratory protocols in protocols.io, where a protocol can be assigned its own identifier (DOI) such that it can be cited independently in the future. For instructions see: http://journals.plos.org/plosone/s/submission-guidelines#loc-laboratory-protocols

We look forward to receiving your revised manuscript.

Kind regards,

Heidar Sharafi

Academic Editor

PLOS ONE

Additional Editor Comments (if provided):

1. In Data Availability, authors included "Yes - all data are fully available without restriction". Is there any dataset they want to share with the readers?

2. The Manuscript Title, PICO, aims, results and discussion should be synchronized to present the work done by the authors. As I included above, these are yet confusing, decrease the quality of evidence presentation. In introduction, there is little about what is the burden (economic, etc.) of HCV duplicate testing in a population with little ongoing risky behaviors. Why the duplicate testing is important? is it a problem? how can it impact the HCV screening? The aim should be expressed clearly considering the work done and the results will be presented.

3. In Abstract, "aggressive screening efforts in high prevalence populations have led to significant costs despite a lack of clinical utility". I am very confused with that statement. WHO recommends screening in populations with high prevalence of HCV (>2% or >5%) such as PWID, FSWs, etc.

4. The methods, patient recruitment and documentation is not clear for the readers. As I get, this is a retrospective study using the data accumulated in EHR. I am not against using such data however, the reader needs to know what exactly happened. Was the screening started before 2012? how the baby boomers linked to diagnostics? what were the diagnostic services (methods, labs, etc)? how the results were documented in EHR? was there any counseling from Labs or clinics after testing? 

5. Were All the retestings duplicates? any triplicates? 

6. I am confused why the authors observed the retesting in those with negative HCV Ab as an appropriate phenomenon while these individuals mostly have no risk for transmission of HCV.

7. What was included in the data of Figure 1? The individuals?, the test results? or the visits in clinic? 

8. What was the number of individuals? number of tests? number of visits in clinic? these can be included in the results.

9. A Table presenting baseline characteristics of patients is needed.

10. In Figure 2, it is not clear what the duplicate tested branch is presenting? the initial testing results or the retesting results.

11. In Table 1, I guess public insurance should have OR more than 1.

12. In Table 2, the second column can be used for presentation of HCV tested more than once and the third column for HCV tested once.

13. 95% CI of OR for gender in Table 2?

Journal Requirements:

2. Thank you for including your ethics statement: "Oversight of this project was provided by the DUCOM Institutional Review Board (Protocol ID # 1702005228). A waiver of consent and waiver of Health Insurance Portability and Accountability Act authorization were provided."

a.Please amend your current ethics statement to include the full name of the ethics committee/institutional review board(s) that approved your specific study and confirm that your named institutional review board or ethics committee specifically approved this study.

b.Once you have amended this/these statement(s) in the Methods section of the manuscript, please add the same text to the “Ethics Statement” field of the submission form (via “Edit Submission”).

3. Please include your tables as part of your main manuscript and remove the individual files. Please note that supplementary tables (should remain/ be uploaded) as separate "supporting information" files

Reviewers' comments:

Reviewer's Responses to Questions

**Comments to the Author**

1. Is the manuscript technically sound, and do the data support the conclusions?

Reviewer #1: Yes

Reviewer #2: Partly

Reviewer #3: Yes

2. Has the statistical analysis been performed appropriately and rigorously? 

Reviewer #1: Yes

Reviewer #2: Yes

Reviewer #3: Yes

3. Have the authors made all data underlying the findings in their manuscript fully available?

Reviewer #1: Yes

Reviewer #2: Yes

Reviewer #3: Yes

4. Is the manuscript presented in an intelligible fashion and written in standard English?

Reviewer #1: Yes

Reviewer #2: Yes

Reviewer #3: Yes

5. Review Comments to the Author

Reviewer #1: Dear editor

Many thanks for the invitation.

I read the manuscript carefully and found it useful and interesting. Overall, the manuscript is well-written. It evaluates two factors; a strategy for screening of Hepatitis C virus (HCV) infection and the rate of duplicated HCV infection tests.

Implementing the strategy has led to increasing the screening rate from 20% to 82%. They’ve found 1027 (18%) duplicate testing among 6717 screened patients. Also, among patients with duplicate tests, 14% had unnecessary testing. Finally, with multivariate analysis, they’ve shown that factors including Age, race, type of insurance and clinic location are associated to duplicate testing.

I only recommend that

1- The authors provide more exact information regarding eligibility criteria and study type in a specific section in the methods part.

2- The authors provide P-value for reported adds ration in the results part.

Best

Rezaee-Zavareh

Reviewer #2: This article is based on these Centers for Disease Control guidelines about screening baby boomers, and the possibility of “duplicate testing”,

It is important to note that at the current time the recommendation is the screen all adults and all pregnant women, this manuscript should be updated to look at both baby boomer “duplicate screening” and potential problems with “duplicate screening” in all adults and all pregnant women

There is a statement in the abstract that there is a lack of clinical utility for screening but this is not correct: utility and cost effectiveness has been proven please see AASLD and CDC guidelines, surveillance is recommended in at risk individuals, yes this is “duplicate” but it follows guidelines since at risk patients must be tested regularly and is not wasteful

There is a clear need for “duplicate testing” or “repeat testing” in at risk individuals who show evidence of high-risk behavior, How did the authors document that there was unnecessary duplicate testing, is there adequate documentation in the EMR that these are low risk individuals with no high-risk behavior?

AST and ALT are liver enzymes and or not liver function tests, were liver function test assessed or liver enzymes assessed?

Justification for repeat screening, surveillance, should not be dependent on elevated liver enzymes but on risk behaviour, what risk assessments to place in this patient population how was that documented

Reviewer #3: It is better that the authors present the data that is dis-aggregated by clinics.

All the analyses should be adjusted for the cluster effect, as there are clusters (clinics) in the data.

A more rigorous conclusion is needed. The conclusion does not support the results. I see that part of duplicate testings were helpful. So, the authors should discuss if the rate of "unnecessary" duplicate testing is practically important.

6. PLOS authors have the option to publish the peer review history of their article (what does this mean?). If published, this will include your full peer review and any attached files.

Reviewer #1: Yes: Mohammad Saeid Rezaee-Zavareh

Reviewer #2: Yes: Robert Gish

Reviewer #3: Yes: Sana Eybpoosh

---

## [Author Response · Author response to Decision Letter 0]

15 May 2020

May 12, 2020

To the Editors of PLoS ONE:

We are resubmitting the manuscript previously titled “Screening Baby Boomers for Hepatitis C in Urban Primary Care Clinics: Does Duplicate Testing Make a Difference?” for further review. Reviewer comments have been considered and the paper has been edited to reflect the recommended revisions. We have itemized the comments and our responses to those comments below.

Editor comments:

Concern: “the aim of the study looks confusing. Reading the manuscript title, I expected that duplicate testing is a new strategy for screening of Hep C patients and the study is going to evaluate this strategy.”

Response: We have changed the title to better reflect the aims of the study.

Concern: “Reading introduction and aim, I thought the duplicate testing is something unexpected in the primary care clinics but it is not elucidated why it cannot be prevented while they have EHR to document the HCV testing.”

Response: The study’s introduction and study aims have been re-written to clarify the goals of the paper. Previously described data regarding duplicate testing in the setting of available EHR records has been cited. 

Concern: “In Data Availability, authors included "Yes - all data are fully available without restriction". Is there any dataset they want to share with the readers?”

Response: We will be sharing our dataset with readers with this submission.

Concern: “The Manuscript Title, PICO, aims, results and discussion should be synchronized to present the work done by the authors. As I included above, these are yet confusing, decrease the quality of evidence presentation. In introduction, there is little about what is the burden (economic, etc.) of HCV duplicate testing in a population with little ongoing risky behaviors. Why the duplicate testing is important? is it a problem? how can it impact the HCV screening? The aim should be expressed clearly considering the work done and the results will be presented.”

Response: The study’s introduction has been re-written to clarify the aims of the study. As the study was not a true interventional study (that is, there is no control group), it has been more correctly framed based on its observational aims. The edited version synchronizes the study’s introduction and aims with the paper’s results and discussion. 

Concern: “In Abstract, "aggressive screening efforts in high prevalence populations have led to significant costs despite a lack of clinical utility". I am very confused with that statement. WHO recommends screening in populations with high prevalence of HCV (>2% or >5%) such as PWID, FSWs, etc.”

Response: The abstract has been re-written to clarify the above as well as the goals of the paper.

Concern: “The methods, patient recruitment and documentation is not clear for the readers. As I get, this is a retrospective study using the data accumulated in EHR. I am not against using such data however, the reader needs to know what exactly happened. Was the screening started before 2012? how the baby boomers linked to diagnostics? what were the diagnostic services (methods, labs, etc)? how the results were documented in EHR? was there any counseling from Labs or clinics after testing?”

Response: The methods section now explains that screening rates were evaluated starting in 2012 as this was the year the new CDC guidelines were implemented. More details were provided regarding the data extraction as structured fields linked to baby boomer data. A statement was added regarding follow-up counseling provided by the C for Cure program.

Concern: “Were All the retestings duplicates? any triplicates?”

Response: There were only three triplicates noted in the study. This has been added to the results section.

Concern: “I am confused why the authors observed the retesting in those with negative HCV Ab as an appropriate phenomenon while these individuals mostly have no risk for transmission of HCV.”

Response: This has been clarified in the Definitions sub-section of the Methods section with a citation to the AASLD-IDSA HCV guidelines. 

Concern: “What was included in the data of Figure 1? The individuals?, the test results? or the visits in clinic?”

Response: The findings in Figure 1 have been clarified in the Results section.

Concern: “What was the number of individuals? number of tests? number of visits in clinic? these can be included in the results.”

Response: The total number of patients and their characteristics are now described in Table 1 and paragraph 1 of the results section. The total number of tests performed is now included in the results section (line 117). We do not have the total number of clinic visits available.

Concern: “A Table presenting baseline characteristics of patients is needed.”

Response: This table has been created and is now Table 1.

Concern: “In Figure 2, it is not clear what the duplicate tested branch is presenting? the initial testing results or the retesting results.”

Response: The branchpoint has now been clarified that the results are of the retesting (duplicate) results.

Concern: “In Table 1, I guess public insurance should have OR more than 1.”

Response: In the univariate analysis predicting ever tested (as shown in what was table 1 but is now table 2) with the same set of subjects as in the multivariate, public insurance does indeed have an OR greater than 1 (1.084) as would be expected since the percentage ever tested is greater in the public group than the private group. However, the analysis in table 2 is multivariate, adjusted for the other variables, and the direction of the association is reversed by the confounder correction. We redid the analyses with the same variables and coding to be sure there was not an accidental switch such that the wrong group was being counted as the reference in the model, and this is not the case. It appears that this small but statistically significant association is tied to other factors. Adding just race or just practice to the model, in addition to insurance type, reverses the direction of the association for insurance type. Adding sex or birth year high/low does not. So, it appears that the association of insurance type with ever tested is confounded by associations with race and practice, which makes sense.

Concern: “In Table 2, the second column can be used for presentation of HCV tested more than once and the third column for HCV tested once.”

Response: We have switched the columns accordingly.

Concern: “95% CI of OR for gender in Table 2?”

Response: The CI of OR for gender has been included.

Reviewer #1:

Concern: The authors provide more exact information regarding eligibility criteria and study type in a specific section in the methods part.

Response: We have clarified the inclusion criteria in the Study Population sub-section. We have added a statement about study design under the new sub-section, Study Design.

Concern: “The authors provide P-value for reported odds ration in the results part.”

Response: P-values are included in our tables reporting odds ratios (now Tables 2 and 3).

Reviewer #2:

Concern: “It is important to note that at the current time the recommendation is the screen all adults and all pregnant women, this manuscript should be updated to look at both baby boomer “duplicate screening” and potential problems with “duplicate screening” in all adults and all pregnant women”

Response: This dataset only includes data for baby boomers. We have now included the expansion of our study to all adults.

Concern: “There is a statement in the abstract that there is a lack of clinical utility for screening but this is not correct: utility and cost effectiveness has been proven please see AASLD and CDC guidelines, surveillance is recommended in at risk individuals, yes this is “duplicate” but it follows guidelines since at risk patients must be tested regularly and is not wasteful

There is a clear need for “duplicate testing” or “repeat testing” in at risk individuals who show evidence of high-risk behavior, How did the authors document that there was unnecessary duplicate testing, is there adequate documentation in the EMR that these are low risk individuals with no high-risk behavior?”

Response: In our limitations section we now address that our study is limited due to lack of knowledge regarding comorbidities that might increase risk. We have labeled patients who have had repeat antibody screening if their first screen is negative as “potentially appropriate” to address this concept within the given limitations of our data.

Concern: “AST and ALT are liver enzymes and or not liver function tests, were liver function test assessed or liver enzymes assessed?”

Response: We did not collect liver function tests as a component of our dataset. However, this is an excellent recommendation for data to be collected in future studies.

Reviewer #3

Concern: “All the analyses should be adjusted for the cluster effect, as there are clusters (clinics) in the data.

Response: Adjusting for clusters in the multivariate analysis is a reasonable suggestion, but we do not believe it quite applies here. If we had randomly selected 5 units out of 20 and wanted to generalize to all 20, we would have to take into account the fact that the 5 might be atypical in some way. For example, if half the 20 units had a positive association between male gender and testing, while the other half had a negative association, we could easily by chance select 5 that showed the positive association, a different 5 that showed the negative, or 5 that were so mixed as to show no association. Adjustment for clustering would address that risk and give more generalizable results.

However, in this study we used essentially all of the units within the system. Our practices are not a sample from a larger citywide or nationwide set. They are a complete set within a single institution. Adjusting for clustering would not help with generalization to the system, since we included all the units (and all relevant patients within each). There is a question of generalization to the broader world, but that is not easily addressed given that this was a single institution study. If we had just taken one large unit from this study and analyzed that, there would be no cluster effect that we could control for, but exactly the same question of generalizability.

So, we believe that our analysis, with practices controlled as a fixed factor, and with recognition that the results of a single institutional study may have limited generalizability if there are important differences between systems, is an acceptable approach.

Concern: “A more rigorous conclusion is needed. The conclusion does not support the results. I see that part of duplicate testings were helpful. So, the authors should discuss if the rate of "unnecessary" duplicate testing is practically important.”

Response: We edited the conclusion to more strongly follow the results. We clarified that providers concerned about ongoing risk in patients who might have been treated or spontaneously cleared might have evaluated that risk more appropriately through viral load testing.

Additional Requirements

Request: “When submitting your revision, we need you to address these additional requirements:

Please ensure that your manuscript meets PLOS ONE's style requirements, including those for file naming. The PLOS ONE style templates can be found at http://www.plosone.org/attachments/PLOSOne_formatting_sample_main_body.pdf and http://www.plosone.org/attachments/PLOSOne_formatting_sample_title_authors_affiliations.pdf”

Response: We have reviewed the guidelines and have re-formatted our paper accordingly.

Request: “Please amend your current ethics statement to include the full name of the ethics committee/institutional review board(s) that approved your specific study and confirm that your named institutional review board or ethics committee specifically approved this study.

Once you have amended this/these statement(s) in the Methods section of the manuscript, please add the same text to the “Ethics Statement” field of the submission form (via “Edit Submission”).”

Response: We have made the changes as requested.

Request: “Please include your tables as part of your main manuscript and remove the individual files. Please note that supplementary tables (should remain/ be uploaded) as separate "supporting information" files”

Response: We have added our tables and figures to the manuscript.

Thank you for your consideration of this re-submission. We hope these revisions help to address your concerns. Please contact me with any further questions.

Sincerely,

Dagan Coppock, MD Thomas Jefferson University

Division of Infectious Diseases

(c) 203-915-7302

(o) 267-503-8845

(f) 215-503-3417

dagan.coppock@jefferson.edu

---

## [Decision Letter · Decision Letter 1]

21 May 2020

PONE-D-20-03702R1

Hepatitis C antibody screening and determinants of duplicate screening in the birth cohort patients of six urban primary care clinics

PLOS ONE

Dear Dr. Coppock,

Thank you for submitting your manuscript to PLOS ONE. After careful consideration, we feel that it has merit but does not fully meet PLOS ONE’s publication criteria as it currently stands. Therefore, we invite you to submit a revised version of the manuscript that addresses the points raised during the review process.

Authors addressed most of the comments and concerns correctly. There are few more comments by the reviewers should be considered by the authors before acceptance of their manuscript for publication in PLOS ONE.

We look forward to receiving your revised manuscript.

Kind regards,

Heidar Sharafi

Academic Editor

PLOS ONE

Reviewers' comments:

Reviewer's Responses to Questions

**Comments to the Author**

1. If the authors have adequately addressed your comments raised in a previous round of review and you feel that this manuscript is now acceptable for publication, you may indicate that here to bypass the “Comments to the Author” section, enter your conflict of interest statement in the “Confidential to Editor” section, and submit your "Accept" recommendation.

Reviewer #1: All comments have been addressed

Reviewer #2: All comments have been addressed

Reviewer #3: (No Response)

2. Is the manuscript technically sound, and do the data support the conclusions?

Reviewer #1: Yes

Reviewer #2: Yes

Reviewer #3: Partly

3. Has the statistical analysis been performed appropriately and rigorously? 

Reviewer #1: Yes

Reviewer #2: Yes

Reviewer #3: No

4. Have the authors made all data underlying the findings in their manuscript fully available?

Reviewer #1: Yes

Reviewer #2: Yes

Reviewer #3: Yes

5. Is the manuscript presented in an intelligible fashion and written in standard English?

Reviewer #1: Yes

Reviewer #2: Yes

Reviewer #3: Yes

6. Review Comments to the Author

Reviewer #1: Dear editor

I checked the revised version of the manuscript together with authors' comments. All of my concerns have been resolved.

Best

Reviewer

Reviewer #2: In the discussion, a small expansion/discussion of the limits of the study due to limited data on risk behaviour or co-morbidities that could lead to repeat testing (not duplicate testing {that implies that the second and third test were not necessary}).

Reviewer #3: The authors have well addressed the comments.

However, for the concern I raised about cluster effect, further attention is needed. In studies where the data is grouped in the form of clusters (here, the clinics), the within-cluster correlations will shrink the total confidence intervals. So, the authors need to correct the confidence-intervals for the "intra-class correlation coefficient". Including all the clinics in your analysis, will not correct the "cluster/design effect", as this issue deals with the correlations that exist within your clusters (i.e., clinics).

7. PLOS authors have the option to publish the peer review history of their article (what does this mean?). If published, this will include your full peer review and any attached files.

Reviewer #1: No

Reviewer #2: Yes: Robert Gish

Reviewer #3: No

---

## [Author Response · Author response to Decision Letter 1]

20 Jun 2020

June 20, 2020

To the Editors of PLoS ONE:

We are resubmitting manuscript PONE-D-20-03702R1, titled “Hepatitis C antibody screening and determinants of initial and duplicate screening in the baby boomer patients of six urban primary care clinics,” for further review. Reviewer comments have been considered and the paper has been edited to reflect the recommended revisions. We have itemized the comments and our responses to those comments below.

Reviewer #1: 

Comment: “I checked the revised version of the manuscript together with authors' comments. All of my concerns have been resolved.”

Response: We thank you for your comments.

Reviewer #2: 

Comment: “In the discussion, a small expansion/discussion of the limits of the study due to limited data on risk behaviour or co-morbidities that could lead to repeat testing (not duplicate testing {that implies that the second and third test were not necessary}).”

Response: Thank you for the recommendation regarding the limitations. In the discussion section, we have expanded on the limits of the study due to our limited dataset. Further, your comments raise excellent points about our terminology. We have changed the language throughout the paper to clarify that duplicate testing may not be “unnecessary,” particularly in the context of ongoing risk/co-morbidities. We have elucidated that repeat antibody screening in patients who already have a baseline positive antibody screen is an inappropriate means for assessing patients with ongoing risk and that viral load testing would be more appropriate. 

Reviewer #3:

Comment: “for the concern I raised about cluster effect, further attention is needed. In studies where the data is grouped in the form of clusters (here, the clinics), the within-cluster correlations will shrink the total confidence intervals. So, the authors need to correct the confidence-intervals for the "intra-class correlation coefficient". Including all the clinics in your analysis, will not correct the "cluster/design effect", as this issue deals with the correlations that exist within your clusters (i.e., clinics).”

Response: This was a very helpful recommendation. We re-performed our regressions, controlling for clustering by clinic. Our odds rations changed and our confidence intervals widened with a decrease in significance for specific variables. We have replaced our previous regression results with the clustered results and have changed our discussion accordingly. 

Thank you for your consideration of this re-submission. We hope these revisions help to address your concerns. Please contact me with any further questions.

Sincerely,

Dagan Coppock, MD 

Thomas Jefferson University

Division of Infectious Diseases

(c) 203-915-7302

(o) 267-503-8845

(f) 215-503-3417

dagan.coppock@jefferson.edu

---

## [Editor Report · Decision Letter 2]

23 Jun 2020

Hepatitis C antibody screening and determinants of initial and duplicate screening in the baby boomer patients of six urban primary care clinics

PONE-D-20-03702R2

Dear Dr. Coppock,

We’re pleased to inform you that your manuscript has been judged scientifically suitable for publication and will be formally accepted for publication once it meets all outstanding technical requirements.

Kind regards,

Heidar Sharafi

Academic Editor

PLOS ONE

---

## [Editor Report · Acceptance letter]

25 Jun 2020

PONE-D-20-03702R2 

Hepatitis C antibody screening and determinants of initial and duplicate screening in the baby boomer patients of six urban primary care clinics 

Dear Dr. Coppock:

I'm pleased to inform you that your manuscript has been deemed suitable for publication in PLOS ONE. Congratulations! Your manuscript is now with our production department. 

Kind regards, 

on behalf of

Dr. Heidar Sharafi 

Academic Editor

PLOS ONE